# Postoperative Femoral Nerve Palsy and Meralgia Paresthetica after Gynecologic Oncologic Surgery

**DOI:** 10.3390/jcm11216242

**Published:** 2022-10-22

**Authors:** Eva Katharina Egger, Oezge Sezer, Mateja Condic, Florian Recker, Milka Marinova, Tobias Hilbert, Arne Koscielny, Alexander Mustea

**Affiliations:** 1Department of Gynecology and Gynecological Oncology, University Hospital Bonn, 53127 Bonn, Germany; 2Department of Nuclear Medicine, University Hospital Bonn, 53127 Bonn, Germany; 3Department of Anesthesiology and Intensive Care Medicine, University Hospital Bonn, 53127 Bonn, Germany; 4Department of Surgery, Elisabeth Hospital, 04277 Leipzig, Germany

**Keywords:** femoral nerve, iatrogenic nerve injury, femoral nerve palsy, meralgia paresthetica

## Abstract

Femoral nerve palsy and meralgia paresthetica following gynecologic cancer surgery are rare, but severe and long lasting. Here, we aimed to study their incidence, severity, possible risk factors and its time to remission. Between January 2008 and December 2017 976 gynecologic cancer patients were identified in our institutional database receiving surgery. Complete patient charts were reviewed retrospectively. Possible risk factors were analyzed by Fisher’s exact test. 441 (45.18%) out 976 were treated for Ovarian cancer. In total 23 patients were identified with a postoperative neurological leg disorder. A femoral nerve palsy was present in 15 patients (1.5%) and a meralgia paresthetica in 8 patients (0.82%). Three patients showed both disorders. Duration of surgery (*p* = 0.0000), positioning during surgery (*p* = 0.0040), femoral artery catheter (*p* = 0.0051), prior chemotherapy (*p* = 0.0007), nicotine abuse (*p* = 0.00456) and prior polyneuropathy (*p* = 0.0181) showed a significant association with a postoperative femoral nerve palsy. Nicotine abuse (*p* = 0.0335) and prior chemotherapy (*p* = 0.0151) were significant for the development of a meralgia paresthetica. Long lasting surgery, patient positioning and femoral arterial catheter placement are risk factors for a postoperative femoral nerve palsy in gynecologic cancer surgery. Polyneuropathy, nicotine abuse, and prior chemotherapy are predisposing risk factors for a femoral nerve palsy and a meralgia paresthetica. A resolution of symptoms is the rule for both disorders within different time schedules.

## 1. Introduction

The femoral nerve is the largest branch of the lumbar plexus and arises from the posterior divisions of the ventral rami of the second, third, and fourth lumbar nerves. These rami form the nerve within the psoas muscle. The nerve rans through the psoas muscle and exits the muscle laterally underneath the fascia of the illiacus muscle about 4 cm above the inguinal ligament. About 1–4 cm caudal of the inguinal ligament the nerve enters the thigh and stays lateral to the femoral vessels. Here the nerve divides into muscular and cutaneous branches supplying the quadriceps muscle, the sartorius muscle, the musculus articularis genus and the skin of the anterior surface of the thigh and the medial surface of the leg and foot [1,2]. A separation of those branches by the lateral circumflex femoral artery is frequent [3]. The nerve shows an increased arterial supply on the right side from the circumflex artery, the iliolumbar and forth lumbar artery [4]. Generally, variations within the anatomic course of the nerve are infrequent [1].

Running outside the gynecologic oncologic operating field, direct injury of the femoral nerve through transsection is rare. However postoperative femoral nerve palsy in gynecologic surgery is possible. Indirect traumas as self-retraining retractors, extreme lithothomy positioning, psoas muscle hematoma and iliac vessel compression after arterial puncture may cause the postoperative nerve palsy [2,5,6,7,8]. Extrapelvic lesions are caused by nerve angulation underneath the inguinal ligament. Extrapelvic lesions show a loss of strength on extension of the leg at knee level, an absent patellar reflex, numbness and/or pain on the medial and anterior aspects of the hip, leg and foot. Intrapelvic lesions, are nerve damages located above the inguinal ligament. They show an additional flexion motor weakness of the hip [1,2,6,9,10]. In general neuropathy rates after major pelvic surgery range in between 0.2–2% [11,12]. In case of a femoral nerve palsy, the sudden postoperative motor deficit disables patients to walk, affecting the postoperative quality of life deeply.

A meralgia paresthetica, a lesion of the lateral cutaneous femoral nerve, followed by numbness and pain of the anterolateral thigh, may be caused by similar mechanisms as a femoral nerve palsy, however this injury may either be underreported or seems to be even less common than a postoperative femoral nerve palsy [13,14]. The lateral cutaneous femoral nerve shows variable anatomical courses. The nerve may originate from the lumbar plexus or from the femoral nerve itself. While traversing the psoas muscle proximally, it is later crossing the illiacus muscle underneath its fascia. The nerve enters the abdominal wall for the most part medial the anterior superior iliac spine. However, other locations as cranial, caudal and lateral to the anterior superior iliac spine, underneath, above or through the inguinal ligament or on the iliac crest have been seen. After entering the thigh it has its individual membranous canal. Further possible variations within the course of the nerve is a division in up to five branches [15,16,17,18].

Here, we aimed to identify reasons and risks as well as predisposing factors for a femoral nerve palsy and a meralgia paresthetica in patients undergoing gynecologic cancer surgery.

## 2. Materials and Methods

Data collection: The study was conducted according to the guidelines of the Declaration of Helsinki and approved by the Ethics Committee of the faculty of Medicine at the university of Bonn, Germany (Nr:427/21). The institutional record database was screened for patients receiving gynecologic oncologic surgery between January 2008 and December 2017. 976 patients were identified. Surgery reports and intraoperative documentation, anaesthesiologic protocols, neurologic consultations and patients charts were reviewed for postoperative documented neurological disorders and their follow up.

Statistical analysis:

All variables were analyzed by Fisher’s exact test to identify significant correlations with a femoral nerve palsy or a meralgia paresthetica. Differences were considered to be significant with *p*-values ≤ 0.05. In case of a meralgia paresthetica a logistic regression analysis was performed. Quantitative variables were summarized using means. All statistical analyses were performed using Minitab Version 18, Minitab LLC., State College, PA, USA.

## 3. Results

### 3.1. Patient Information

#### 3.1.1. General Information

A total of 976 patients received gynecologic oncologic surgery between January 2008 and December 2017 at our institution. The median age was 59 years. The majority of patients (*n* = 441) received surgery for ovarian cancer. The first neurological evaluation by the attending surgeons identified 42 cases with a postoperative leg disorder. In 22 patients, epidural anesthesia was discontinued and re-evaluation was performed several hours later. The other 20 patients had no peridural anesthesia. 23 out of 42 patients required a neurologic consultation due to persisting symptoms and a suspected nerve palsy. 15 patients were diagnosed with a femoral nerve palsy. 3 out of these 15 patients showed an additional meralgia paresthetica. The other eight patients were diagnosed with a meralgia paresthetica only.

#### 3.1.2. Femoral Nerve Palsy

In eight patients the femoral nerve palsy was left sided and in seven patients the femoral nerve palsy was right sided. 13 patients showed an intrapelvic lesion and two patients with vaginal access had an extrapelvic lesion. Ten patients underwent surgery for ovarian cancer. Two patients for endometrial cancer, one patient for vaginal cancer, one patient for vulvar cancer and one patient for cervical cancer. In 2 patients no retractor was used due to a vaginal access, in 9 patients the condor^®^ MedTec retractor was used and in further 4 patients the muenster aesculap^®^ retractor was used. The median BMI was 24 (range 19–30). Analyzed risk factors are shown in Table 1. All oncologic diagnoses are depicted in Table 2.

#### 3.1.3. Meralgia Paresthetica

A meralgia paresthetica was seen in 8 ovarian cancer patients, in one endometrial cancer patient, in one vulvar cancer patient and in one cervical cancer patient. The median age was 49 years (range 42–77), the median BMI was 26 (range 19–33) and the median duration of surgery was 316 min (range 119–605 min). Table 3 summarizes surgical details of all 15 patients with a femoral nerve injury. Table 4 summarizes surgical details of the patients with a meralgia paresthetica. 

#### 3.1.4. Identified Risk Factors

The duration of surgery, the positioning during surgery—candy canes versus maquet boots, the use of an arterial femoral catheter, a history of smoking, prior chemotherapy and a pre-existing polyneuropathy were significant risk factors for a postoperative femoral nerve palsy as seen in Table 5. Significant risk factors for a meralgia paresthetica were prior chemotherapy and nicotine abuse as also seen in Table 5.

#### 3.1.5. Resolution of Symptoms

14 out of 15 patients with a femoral nerve palsy showed a complete resolution of symptoms, within 14 days to 9 months. One patient was lost to follow up. The last visit at 3 months after surgery documented a partial resolution of symptoms. In all cases intense physiotherapy was implemented for recovery. All patients (*n* = 12) without symptom resolution at discharge from hospital were equipped with knee braces, crutches and physiotherapy prescriptions. 11 patients were diagnosed with a meralgia paresthetica. All experienced a complete resolution of symptoms in between 10 days to 2 months. No specific therapy was performed in these patients.

## 4. Discussion

Due to sudden postoperative severe long lasting motor deficits, a femoral nerve palsy affects the quality of life deeply during the rehabilitation process after gynecologic oncologic surgery. In general the patients suffer from neuropraxia, a focal demyelinisation caused by compression or even from axonotmesis, an axonal damage, caused by nerv traction or steep compression [19]. In both cases a restoration of all motor functions is possible and rather the rule as, in contrast to a nerve transection [20]. As the femoral nerve runs outside the gynecologic oncologic operating field, a nerve transection is unlikely. Figure 1 shows the anatomical route of the nerve on CT scans. Intense rehabilitation and physiotherapy are usually mandatory to restore the muscular function and prevent muscle wasting.

Self retaining retractors were early identified as risk factor for a postoperative femoral nerve palsy in gynecologic surgery. Hand held systems were proposed as the blades may press the nerve to the pelvic side wall or even directly affect the nerve. Especially lateral extended transverse incisions were identified to increase the risk for a femoral neve palsy as they lead to an extreme lateral placement of the retractor blades [2,5]. In radical gynecologic oncologic surgery good exposition is essential to free the abdomen from tumor. In order to reach this goal self retaining retractors are mandatory. As a first risk reducing step, the blades in our condor^®^ MedTec retractor system have been individually reshaped as seen in Figure 2.

In contrast to our study, other groups reported that the use of candy canes compared to the use of boot stirrups are at greater risk for a postoperative neuropathy in gynecologic surgery, proposing a better placement in boot stirrups [21,22]. As our finding was contrary to the recent literature and boots stirrups had been implemented as standard positioning device in our department since several years, repeated intraoperative positioning controls were implemented as a standard operating procedure in our department. Boot stirrups allowed an exceptional good exposition for the colorectal anastomosis by an extreme hip flexion. In case of hip extension after the anastomosis, an unintentional persistent external rotation was seen quite frequently. In fact 8 out of 9 patients with a femoral nerve injury with boot stirrups had received a colorectal anasotomosis in our cohort. In obese patients this problem has been observed in laparoscopic colorectal surgery [23]. Lithotomy positioning with extreme flexion, abduction and external rotation of the hip leads to a nerve angulation underneath the inguinal ligament. Depending on the duration of the angulation a postoperative femoral nerve palsy is possible [6,23]. In general the axis from the contralateral shoulder to the lower extremity should be in line with the umbilicus to prevent excessive external hip rotations [24].

In lymphonodectomy femoral nerve injury may be caused by traction, compression and ischemia. In pelvic lymphonodectomy the greatest traction is needed for the exposition near the vena circumflexa ilium profunda [6]. Therefore maintaining maximal traction during lymphonodectomy only and a routine release of the blades after lymphonodectomy seems mandatory [2]. As the blood supply for the fermoral nerve is sparse, originating from small branches of the internal and external iliac artery, it is at risk of ischemia especially in case of branch clamping in pelvic and aortic surgery when bulky lymph nodes are resected [4,12]. 11 out of 15 patients with a femoral nerve palsy had received a lymphonodectomy in our cohort without being significant (*p* = 0.1990).

Ischemia may also be caused by iliac muscle hematomas as the overlaying fascias are robust enough to entrap a hematoma. Patients with a hemorrhagic diasthesis are at risk for this type of ischemia. Postoperative MRI diagnostics are important in these patients to identify the hematoma in order to clear the nerve from pressure for a quick resolution of symptoms [25].

We identified femoral artery catheters, used for routine blood pressure measurement during oncologic surgery, as a significant risk factor for a femoral nerve palsy in our cohort. Femoral artery catheters may cause a direct trauma and bear the risk of a nerve compression by hematoma and aneurysm formation. They are easily replaced by radial artery catheters [9,26].

The taxane induced peripheral polyneuropathy is a well-studied adverse side effect of chemotherapy, especially in case of paclitaxel. Smoking is also considered as risk factor for the development of a polyneuropathy [27,28]. To date we could not identify any published evidence regarding the incidence of a femoral nerve palsy in relation to a preexsistant polyneuropathy, preoperative chemotherapy or smoking. All three preexsistant factors showed a significant correlation with a postoperative femoral nerve palsy in our cohort.

The general incidence of a meralgia paresthetica is 4.3 per 10,000 person years [29]. While it has been linked to pregnancy, diabetes mellitus, obesity, tight seat belts, intraabdominal masses, tight trousers, orthopedic and laparoscopic surgeries, reports considering gynecologic oncologic surgery are scarce [13,14]. To our knowledge we are the first to report a significant association of nicotine abuse and preoperative chemotherapy in gynecologic oncologic surgery and meralgia paresthetica.

Femoral nerve injury is seen more frequently in case of lateral transpsoas lumbar interbody fusion surgery [30]. The implementation of intraoperative femoral nerve monitoring proofed to be a successful prevention tool to avoid this injury. As soon as reduced femoral nerve evoked potentials were recorded during surgery, surgeons and anesthesiologists promptly took simple actions as removing surgical retraction or trying to increase the blood pressure [31]. Those countermeasures led to a full recovery of the decreased evoked potential amplitudes without any sign of a postoperative femoral nerve injury. Waiving those simple actions, patients experienced postoperative femoral nerve injuries. Making this an interesting tool for other surgical interventions as well [32].

There are limitations to our study to mention, as the retrospective design and the low number of patients affected, increasing the risk of overseeing further risk factors.

Nevertheless, we identified risk factors which may easily be ruled out in gynecologic oncologic surgery as femoral artery catheters and patient positioning. Following our analysis several standard operating procedures have been implemented in our daily routine of gynecologic oncologic surgery: a change of retractor blades as seen in Figure 2. In case of ovarian cancer, routine surgery starts in the upper abdomen to decrease the time of an unintentional unfavorable leg placement in boot stirrups. Leg positioning is routinely controlled after colorectal anastomoses. Femoral artery catheters have been replaced by radial artery catheters. Preexisting factors as neoadjuvant chemotherapy, smoking and a polyneuropathy cannot be ruled out, but they may serve as special reminder for above mentioned standard operating procedures in oncologic surgeries in order to minimize postoperative neuropathies.

## 5. Conclusions

Long lasting surgery, patient positioning and femoral arterial catheter placement are risk factors for a postoperative femoral nerve palsy in gynecologic oncologic surgery. Polyneuropathy, nicotine abuse, and prior chemotherapy are predisposing risk factors for a femoral nerve palsy and a meralgia paresthetica, which should be acknowledged prior to gynecologic oncologic surgery. A resolution of a postoperative femoral nerve palsy and a meralgia paresthetica is the rule. However, symptoms may persist up to nine months in case of a femoral nerve palsy.

## Figures and Tables

**Figure 1 jcm-11-06242-f001:**
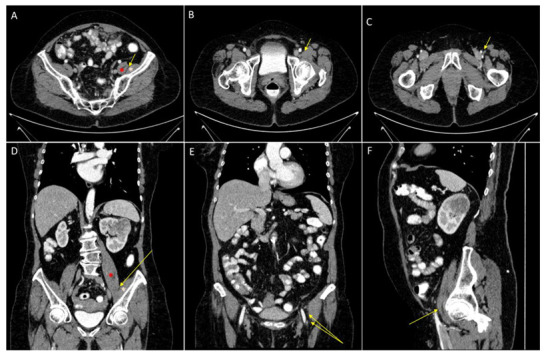
Anatomical route of the femoral nerve. Femoral nerve (yellow arrow), psoas muscle (red star) and illiacus muscle (pink star). A/B/C axial contrast-enhanced CT scans showing the course of the femoral nerve in between the psoas and iliacus muscle (**A**) and running lateral to the femoral artery and vein (**B**,**C**). (**D**,**E**) coronal contrast-enhanced CT scans showing the course of the femoral nerve in between the illiacus muscle and the psoas muscle and lateral to the fermoral artery and vein.(**F**) saggital contrast-enhanced CT scans showing the nerve just below the inguinal ligament.

**Figure 2 jcm-11-06242-f002:**
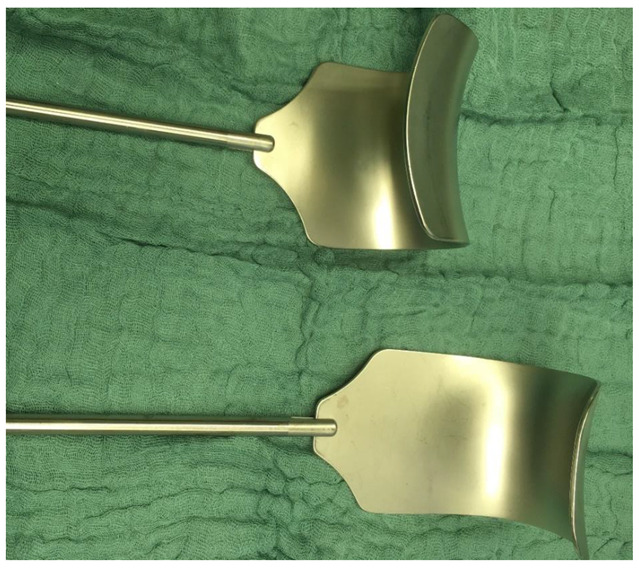
Retractor blades; cranial: old blades; caudal: individually reshaped new blades.

**Table 1 jcm-11-06242-t001:** Risk factors.

	Femoral Nerve Injury	Femoral Nerve Intact
Femoral artery catheter	Yes: 5 No: 10	Yes: 75 No: 886
Peridrual catheter	Yes: 8 No: 7	Yes: 329 No: 632
Duration of surgery in minutes	Median 413 (105–624)	Median 243 (15–770)
Lymphnode dissection	Yes:11 No: 4	Yes: 535 No: 426
Positioning	Candy canes: 6 Boot stirrups: 9	Candy canes: 725 Boot stirrups: 236
Age	Median 65 (42–75)	Median 59 (14–88)
Arterial hypertension	Yes: 5 No: 10	Yes: 353 No: 608
Smoking	Yes: 4 No:11	Yes: 89 No:872
Polyneuropathy	Yes:2 No: 13	Yes: 13 No: 948
Chemotherapy before surgery	Yes: 10 No: 5	Yes: 234 No: 727
Coronary heart disease	Yes: 0 No: 15	Yes: 45 No: 916
Diabetes mellitus	Yes: 0 No:15	Yes: 112 No: 849
BMI <185 18.5–24.9 25–29.9 30–34.9 35–39.9 >40 n/a	0 8 4 3 0 0 0	20 404 244 139 37 56 61

n/a: not available.

**Table 2 jcm-11-06242-t002:** Number of patients and corresponding diagnoses.

Entity	Number of Patients	Percent (%)
Cervical cancer in total	117	11.9
primary adenocarcinoma	25	2.6
primary squamous carcinoma	77	7.9
recurrent adenocarcinoma	2	0.2
recurrent squamous carcinoma	11	1.1
mucinous carcinoma	2	0.2
Ovarian cancer in total	441	45.2
Primary	289	29.6
Recurrent	79	8.1
borderline tumor	73	7.5
Vulvar cancer in total	191	19.6
Primary	146	14.9
Recurrent	41	4.2
Basalioma	4	0.4
Vulvar melanoma	30	3.1
Endometrial cancer in total	148	15.2
Primary	127	13.0
Recurrent	7	0.7
Carcinosarcoma	14	1.4
Uterine sarcoma	11	1.1
Stromal	9	0.9
Leiomyosarcoma	2	0.2
Vaginal carcinoma	8	0.8
Primary	6	0.6
Recurrent	2	0.2
Sex cord stromal tumors	16	1.6
Granulosacell tumor	10	1.0
Recurrent granulosacell tumor	5	0.5
Sertoli-Leydig cell tumor	1	0.1
Germ cell tumors	8	0.8
Dysgerminoma	1	0.1
Malign teratoma	3	0.3
Yolk sac tumor	2	0.2
Embryonalcell tumor	1	0.1
Chorioncarcinoma	1	0.1
Krukenberg Tumor	3	0.3
Mesothelioma	1	0.1
Carcinoid	1	0.1
Sweat gland carcinoma	1	0.1

**Table 3 jcm-11-06242-t003:** 15 patients with femoral nerve injury.

Age	Diagnosis	Surgery	Surgery in Min	Patient Positioning	Retractor	BMI	Medical History 1. Prior Chemotherapy 2. Nicotine Abuse 3. PNP
65	OC	CS, pp LNE	511	Li, B	C	25	1, 3
43	OC	CS, pp LNE	388	Li, CC	C	30	1
59	VaC	VV, ing. LNE	105	Li, B	None	25	2
67	EC	CS, pp LNE	581	Li, B	C	19	1, 3
75	OC	CS	456	Li, B	C	21	1
71	OC	CS, ppLNE	413	Li, B	M	24	1
74	OC	CS, pp LNE	430	Li, CC	C	21	None
42	OC	CS, ppLNE	601	Li, B	C	27	1, 2
75	OC	CS	193	Li, CC	M	24	None
71	VC	VV, ing. LNE	175	Li, CC	None	30	None
65	EC	L-HE-AE-OE, pp LNE	510	Li, B	C	20	1
70	CC	RHE	305	Li, CC	M	30	None
59	OC	CS, ppLNE	524	Li, B	C	19	1
45	OC	CS, pp LNE	624	Li, B	C	27	2
48	OC	CS, pp LNE	345	Li, CC	M	22	1, 2

**Table 4 jcm-11-06242-t004:** 11 patients with a meralgia paresthetica.

Age	Entity	Surgery	Surgery in Min	Positioning	Retractor	BMI	Comorbidity 1. After Chemotherapy 2. Nicotine Abuse
72	OC	CS, pp LNE	605	Li, B	C	26	None
71	VC	VV; ing LNE	175	Li, CC	None	30	None
70	EC	CS	295	Li, CC	M	33	None
53	OC	CS, ppLNE	365	Li, B	C	27	1
50	OC	CS, ppLNE	534	Li, CC	C	26	1
49	OC	CS	440	Li, CC	C	26	1
49	OC	CS, pp LNE	260	Li, CC	M	32	1
48	CC	HE—AE	119	Li, B	C	19	2
48	OC	CS, pp LNE	345	Li, CC	M	22	1, 2
46	OC	CS	316	Li, CC	M	20	None
42	OC	CS pp LNE	601	Li, B	C	27	1, 2

OC: ovarian cancer, EC: endometrial cancer, VC: vulvar cancer, VaC: vaginal cancer CC: cervical cancer, Dys: dysgerminom, Li: lithtomy position, B: stirrup boots, CC: candy canes, C: condor© retractor, M: muenster aesculap© retractor, CS: cytoreductive surgery, pp LNE: pelvic and paraaortic lymphnode dissection, ing. LNE: inguinal groin resection, RHE: radical hysterectomy and pelvic lymphondektomy, L-HE-AE-OE: laparotomy + hysterectomy + oophorectomy + omentectomy, HE-AE: hysterectomy and oophorectomy VV: vulvectomy.

**Table 5 jcm-11-06242-t005:** Significant correlations by Fisher`s exact test for a femoral nerve injury and a meralgia paresthetica.

	Femoral Nerv Palsy	
Factor		*p*-Value
Duration of surgery	3/447 < 245 min 12/529 > 245 min	0.0000
Positioning	9/245 with boot stirrups 6/731 with candy canes	0.0040
Arterial femoral catheter	5/80 with catheter 10/896 without catheter	0.0051
Prior chemotherapy (ctx)	10/244 with ctx 5/732 without ctx	0.0007
Nicotine abuse (na)	4/93 with na 10/883 without na	0.0456
Prior polyneuropahty (pp)	2/16 with pp 13/960 without pp	0.0181
	Meralgia Paresthetica	
Factor		*p*-Value
Nicotine abuse (na)	4/93 with na 7/883 without na	0.0335
Prior chemotherapy (ctx)	6/244 with ctx 5/732 without ctx	0.0151

## Data Availability

All data involved in this study will be made available by the corresponding author upon request.

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
