# Peer review of "Postoperative Femoral Nerve Palsy and Meralgia Paresthetica after Gynecologic Oncologic Surgery"

_jcm, 2022, doi:10.3390/jcm11216242_

Round 1

Reviewer 1 Report

The authors present a study where they looked at causal factors for postoperative femoral and lateral femoral cutaneous nerve injuries following gynecologic surgeries.

The authors should consult someone with neuroanatomy experience to assist them in writing about the nerves they are discussing. They are grossly inaccurate in the description provided. 

Introduction –The entire anatomical course for the femoral nerve the authors present is incorrect and needs to be addressed.  The femoral nerve forms within the substance of the psoas muscle and traverses within the psoas muscle until it exits from the lower lateral border of the psoas.

When discussing meralgia paresthetica – the authors state it is a lesion of “the lateral cutaneous nerve of the thigh” – This is incorrect the lateral FEMORAL cutaneous nerve is the nerve that is lesioned. The Lateral femoral cutaneous nerve is a separate and distinct nerve from the femoral nerve and it should be delineated as such in the manuscript.  The LFCN arises from the L2 nerve root within the substance of the psoas muscle as well. The authors should highlight the differences in course and anatomy between the 2 nerves.

The authors should add a section about the use of neuromonitoring to mitigate postoperative femoral nerve injury. There have been many papers showing the benefits of neuromonitoring of the femoral nerve during transpsoas approaches to the spine. The papers have shown very high specificity and PPV with femoral nerve monitoring techniques. The authors can start with the following citation

Silverstein JW, Block J, Smith ML, et al. Femoral nerve neuromonitoring for lateral lumbar interbody fusion surgery. Spine J. 2022;22(2):296-304. doi:10.1016/j.spinee.2021.07.017

Author Response

Dear Reviewer,

thanks for your helpful comments.

The follwing revisions were mede within the manuscript according to your suggestions:

  1. the anatomical course of the femoral nerve has been reviesed in total. This is marked in red from line 33 to 44
  2. the antomical route of the lateral cutaneous femoral nerve has been revised and extended. Line 59-70, marked in red
  3. Neuromonitoring is discussed within the dicussion section from line 221-229, marked in red.

Reviewer 2 Report

The article is about an interesting subject and well written. The data give significant indication for the surgeons. The Authors report that they changed their procedures in the operative room. This change happened after they made this retrospective analysis of their case series? Does that reduced complications?

Author Response

Dear Reviewer,

Thank your for your review. We did change our SOP`s after our data evaluation. So far we have had only one more case within 18 month of a femoral nerve injury. But due to the rarity of the injury we did not feel that this is representative. 

Kind regards